# Imaging of Tauopathies with PET Ligands: State of the Art and Future Outlook

**DOI:** 10.3390/diagnostics13101682

**Published:** 2023-05-09

**Authors:** Miriam Conte, Maria Silvia De Feo, Marko Magdi Abdou Sidrak, Ferdinando Corica, Joana Gorica, Giorgia Maria Granese, Luca Filippi, Giuseppe De Vincentis, Viviana Frantellizzi

**Affiliations:** 1Department of Radiological Sciences, Oncology and Anatomo-Pathology, Sapienza University of Rome, 00161 Rome, Italy; 2Department of Nuclear Medicine, Santa Maria Goretti Hospital, 00410 Latina, Italy

**Keywords:** tauopathies, Alzheimer’s disease, PET imaging, tau PET, TSPO, PiB, MAPT, amyloid plaque

## Abstract

(1) Background: Tauopathies are a group of diseases characterized by the deposition of abnormal tau protein. They are distinguished into 3R, 4R, and 3R/4R tauopathies and also include Alzheimer’s disease (AD) and chronic traumatic encephalopathy (CTE). Positron emission tomography (PET) imaging represents a pivotal instrument to guide clinicians. This systematic review aims to summarize the current and novel PET tracers. (2) Methods: Literature research was conducted on Pubmed, Scopus, Medline, Central, and the Web of Science using the query “pet ligands” and “tauopathies”. Articles published from January 2018 to 9 February, 2023, were searched. Only studies on the development of novel PET radiotracers for imaging in tauopathies or comparative studies between existing PET tracers were included. (3) Results: A total of 126 articles were found, as follows: 96 were identified from PubMed, 27 from Scopus, one on Central, two on Medline, and zero on the Web of Science. Twenty-four duplicated works were excluded, and 63 articles did not satisfy the inclusion criteria. The remaining 40 articles were included for quality assessment. (4) Conclusions: PET imaging represents a valid instrument capable of helping clinicians in diagnosis, but it is not always perfect in differential diagnosis, even if further investigations on humans for novel promising ligands are needed.

## 1. Introduction

Tauopathies are a complex collection of clinical syndromes characterized by the deposition of misfolded tau proteins (tubulin-associated unit, also called microtubule-associated protein tau (MAPT) protein), chiefly in neurons but also in glial cells and the extracellular space [1,2]. They include Pick’s disease (PiD), progressive supranuclear palsy (PSP), corticobasal degeneration (CBD), argyrophilic grain disease, globular glial tauopathies (GGT), and primary age-related tauopathies, such as neurofibrillary tangle dementia, chronic traumatic encephalopathy (CTE), and aging-related tau astrogliopathy [2]. Generally, they can be classified on the basis of tau isoform in 3R tauopathies with a principal isoform 3R, 4R tauopathies (mainly having a 4R tau isoform), or 3R/4R tauopathies when 3R and 4R give the same contribution. In some diseases, tau protein aggregation is secondary to other protein misfoldings, such as amyloid beta (Aβ) in Alzheimer’s disease (AD), the second cause of dementia in the world [3,4,5], and repetitive brain injury in CTE [6]. Even if certainty of diagnosis is obtained in post-mortem examinations [7], the complexity of clinical presentation requires a more efficient and specific diagnostic evaluation that could guide clinical diagnosis and therapeutic decisions. In this field, PET (positron emission tomography) imaging represents an emerging and in-evolution instrument capable of facing this challenge, from diagnosis to prediction of cognitive decline and outcome evaluations [8]. In this systematic review, the actual and emerging PET tracers for tauopathies were examined and discussed.

## 2. Materials and Methods

### 2.1. Search Strategy and Study Selection

This systematic review was realized following PRISMA guidelines [9]. An online literature search was made on different databases: Pubmed, Scopus, Medline, Central (Cochrane Library), and the Web of Science. Articles published from January 2018 to 9 February 2023, were searched. The chosen keywords applied in each database query were “pet ligands” and “tauopathies”. The eligible studies explored the development of novel PET radiotracers for imaging in tauopathies or were comparative studies between existing PET tracers. Reviews, case reports, studies on machine learning, the synthesis of cold molecules, ongoing trials, absent full-texts, and articles not related to the subject of research were excluded. The English language was mandatory.

### 2.2. Quality of the Selected Studies

For each article, general data were retrieved, such as authors, journal, year of publication, country, and study design. The selected studies were evaluated through the Quality Assessment of Diagnostic Accuracy Studies-2 (QUADAS2) tool. Data extraction and quality assessment were separately executed by two reviewers, and any disagreements were resolved by discussion among the authors.

## 3. Results

### 3.1. Search Results

The research produced a total of 126 articles. Of these, 96 articles were identified from PubMed, 27 from Scopus, one from Central, and two from Medline. No studies were collected from the Web of Science. Twenty-four duplicate records were removed. A total of 102 articles with potential relevance were assessed for eligibility through the examination of each abstract. From the overall group of 102 papers, 63 articles were excluded because they did not satisfy the inclusion criteria. Among them, 25 were reviews, six were case reports, one paper was an article in Portuguese, one study concerned machine learning, three articles were related to the synthesis of cold ligands but no radiolabeling was conducted, one paper had no available full-text, one study was an ongoing trial, and 25 papers were not related to the subject of interest. The remaining 39 articles were included and selected for quality assessment. Figure 1 illustrates the flow chart for the representation of the search strategy and selection criteria.

### 3.2. Study Characteristics

The main thematic areas of the selected studies could be summarized as follows: (1) studies on microglia activation; (2) studies about PET tracers for specific tau filaments; (3) studies about novel tracers for protofibrils; (4) comparative studies between amyloid tracers; (5) distribution’ studies of tau ligands; (6) studies on novel ligands; (7) studies concerning important genes involved in neurodegeneration. Between them, ten articles were on microglia activation, four were concerned with ligands for specific tau filaments, two were on protofibril tracers, three were comparative studies between amyloid tracers, five were distribution studies on tau ligands, nine were on novel ligands, and six were studies about genes involved in neurodegeneration.

### 3.3. Methodological Quality Assessment

The methodological quality of the papers included in the study was very high since they satisfied all the QUADAS-2 domains. The selected studies raised low concerns about bias and applicability. The lack of autoptic examinations and the low numerosity of the population in some studies were not considered the reason for bias because of the rarity of some explored tauopathies and the impossibility to conduct autoptic exams in living subjects.

## 4. Discussion

### 4.1. Studies on Microglia Activation

Hallmarks of Alzheimer’s disease are the presence of extracellular Aβ (amyloid β) plaques (also called neuritic or senile plaques), composed of Aβ protein aggregates, and intraneuronal aggregated hyperphosphorylated tau protein inclusions, identified as a subsequent process [10]. Inflammation also plays a key role in pathogenesis since it appears to be the leading phenomenon that precedes amyloid deposition. In this scenario, the microglia of cerebral phagocytic cells, which cause the primary immune response of the central nervous system, prevent amyloid plaque formation [11]. However, at the beginning of neuronal degeneration, they are stimulated to produce pro-inflammatory mediators, which are co-actors in the neuronal dysfunction process, while the recruitment of astrocytes carries over the inflammation [12]. As we age, microglia become more sensitive to stimuli and less efficient at removing damaged cells, which themselves become paradoxically potent inflammatory triggers. In other words, the microglia function impairment behaves as the cause of further neuronal damage [13]. Malpetti and colleagues examined the relationship between tau pathology measured with ^18^F-AV-1451 (also known as ^18^F-flortaucipir or [^18^F]T807) PET scan, neuroinflammation detected by ^11^C-PK11195 PET imaging, brain atrophy studied with MRI, and longitudinal cognitive changes over 3 years in 12 patients with a clinical diagnosis of probable Alzheimer’s disease and 14 with a positive amyloid PET imaging amnestic MCI [14]. In particular, ^11^C-PK11195 was chosen as a tracer since it is capable of binding the 18-kDa translocator protein (TSPO), a protein of the outer mitochondrial membrane whose expression is upregulated in activated microglia [15]. They demonstrated the combined and independent value of the three imaging methods in predicting cognitive decline. Bayesian multiple regression with brain components and demographic variables demonstrated the superiority of PET as a predictor in contrast to atrophy. Tau burden was identified in the posterior cortical region, while inflammation seemed to involve the anterior temporal lobe, acting both as predictors of cognitive decline.

The research of Rauchmann et al., indeed, was conducted on 32 Aβ-positive early AD subjects and 18 healthy controls who underwent [^18^F]GE-180 PET scans, a third-generation tracer for the detection of TSPO, and MRI imaging [16]. Since the affinity of [^18^F]GE-180 is influenced by the polymorphism rs6971 (Ala/Thr) of the TSPO gene, only high-affinity binders (HABs) and mixed-affinity binders (MABs) were studied. A PET scan was conducted after the intravenous administration of 189  ±  12 MBq [^18^F]GE-180 through dynamic acquisition over 90 min and static images with a time frame of 60–80 min. A PET scan was co-registered with a T1-weighted MRI scan. Increased tracer uptake was seen in the bilateral anterior medial temporal lobe, with the highest mean TSPO-PET uptake in the occipital region. They also found a quadratic association between anterior medial temporal lobe TSPO-PET standardized uptake value ratios (SUVR) and cognitive performance. This implies an underlying activation of the brain’s immune system at that AD stage. From network connectivity evaluations, it was seen that microglia activation was distributed along highly connected brain regions, and the author suggested that AD propagates along the central nervous system via vulnerable connectivity pathways. From these two studies, it is evident that PET imaging represents a valid method in AD imaging, capable of adding more information compared to MRI. Taking into consideration high- or mixed-affinity binders for TSPO PET is a crucial step in dealing with the issue of TSPO gene polymorphism.

Fairley and colleagues demonstrated the protective role of the TSPO ligand Ro5-4864 (Ro5) in rTg4510 tau transgenic mice (TauTg) [17]. MRI scans, tau-PET (^11^C-PBB3), and TSPO-PET (using ^18^F-FEBMP) acquisitions were conducted, and immunochemistry for markers of neuronal survival (NeuN), tauopathy (AT8), and inflammation (TSPO, ionized calcium-binding adaptor molecule 1 or IBA-1, and complement component 1q or C1q) were searched in mice brain sections. Ro5 treatment diminished brain atrophy and the neuronal loss of the hippocampus, but no effect was seen on tau deposition. Neuronal loss and atrophy were linked to inflammation as revealed by TSPO-PET, IBA-1, and C1q levels, and in vitro studies showed how Ro5 lowered C1q expression in activated microglia. In the field of inflammation research, it ranks frontotemporal dementia (FTD), a neurodegenerative disorder that can be distinguished into the behavioral variant (bvFTD), the non-fluent variant primary progressive aphasia (nfvPPA), and the semantic variant primary progressive aphasia (svPPA) [7]. Since the aggregation and misfolding of proteins are pivotal post-mortem criteria for FTD [18], different pathological proteins have been studied, such as tau protein or TAR DNA-binding protein 43 (TDP-43) type C neuropathology in svPPA [19,20], to help the diagnosis of this disease. In the paper of Bevan-Jones et al., they hypothesized that major inflammation and protein aggregation could be revealed in the frontotemporal regions of patients with bvFTD, svPPA, and nfvPPA [21]. The researchers conducted an ^11^C-PK-11195 PET scan and an ^18^F-AV-1451 PET evaluation to detect areas with overexpression of TSPO and with aggregated non-amyloid-β pathological proteins, respectively, assuming the binding of ^18^F-AV-1451 in tauopathies and TDP-43-related disease, even if with lower affinity than in AD. They also thought neuroinflammation and aggregation could have specific distribution patterns that could help in the differential diagnosis. The results confirmed a temporal increase in brain uptake in svPPA and frontotemporal cortex uptake in bvFTD, confirmed at the post-mortem immunohistochemistry exam, with an association between inflammation and protein aggregation. In particular, they found a higher uptake of ^11^C-PK-11195 in patients with bvFTD and the C9ORF72 mutation, while intermediate uptake was seen in patients with bvFTD and abnormalities in MAPT carriers. ^18^F-AV-1451 was investigated in another study by the group of Bevan-Jones and colleagues, which analyzed the binding in bvFTD due to a hexanucleotide repeat expansion in C9orf72, associated with TDP-43 pathology. In particular, the C9orf72 mutation increases binding in the frontotemporal cortex [22]. It is evident that inflammation plays a main role in neuronal loss, and maybe the use of a TSPO PET represents a valid choice in the detection and treatment response evaluation of tauopathies. However, ^18^F-AV-1451 could not be the first choice for AD studies because of its low affinity.

An interesting comparative study between different TSPO tracers has been executed by Ji et al. [23]. Since the expression of TSPO in cerebral blood vessels could interfere with neuroinflammation detection, they conducted in vivo and in vitro autoradiographic imaging of normal and TSPO-deficient mice models using ^18^F-FEBMP, ^11^C-PK11195, ^11^C-PBR28, ^11^C-Ac5216, and ^18^F-FEDAA1106. In normal mice, ^11^C-PK11195, ^18^F-FEDAA1106, and ^11^C-PBR28 had a specific cerebral vessel uptake not seen with ^18^F-FEBMP and ^11^C-Ac5216. They discovered the major contrast of ^18^F-FEBMP in the tau rodent model compared to ^11^C-PK11195.

Despite the fact that TSPO could be helpful in the differential diagnosis between tauopathies, the low cellular specificity and the impact of genotype on uptake [24] led the research to focus on newer targets, as the paper of Horti et al., demonstrated [25]. The study concerns the development of a new PET tracer using the macrophage colony-stimulating factor 1 receptor (CSF1R) inhibitor (5-cyano-N-(4-(4-methylpiperazin-1-yl)-2-(piperidin-1-yl)phenyl)furan-2-carboxamide) radiolabeled with ^11^C ([^11^C]CPPC) and evaluation of it for PET imaging in neuroinflammation. CSF1R (or c-FMS, CD-115, or M-CSFR) is a microglial surface type III tyrosine kinase receptor with low expression in other cells and activated by CSF1 and IL-34 [26,27,28]. CSF1R functions include the regulation of proliferation, differentiation, and survival of hematopoietic precursor cells [29], but they also play a role in microglial development and neuroinflammation [30,31,32,33,34]. However, the “limit” of this molecule, as cited by the authors, is the fact that no healthy mammalian brain studies have been performed in detail. In mice, high levels of CSF1R have been detected in the superior cortical regions and lower levels in the other nervous regions [35]. [^11^C]CPPC demonstrated a specific brain uptake in the murine model, with a peak value of 150% SUV (percentage of standardized uptake value) in the frontal cortex after 5–15 min from the injection and a stability of 150% SUV between 30 and 60 min. The compound demonstrated the specificity of binding for CSF1R in the healthy murine brain, with a relevantly higher uptake in LPS mice and AD-bearing mice. After the administration of LPS to baboons, the tracer uptake increased twice as much. It was fully inhibited by non-radiolabeled CPPC injection, demonstrating a specific binding to CSF1R. The same experiment has been conducted in mice, leading to the same conclusions. The CSF1R receptor was also the target of the novel [^11^C]NCGG401 [36]. This tracer was developed as an imaging biomarker in AD, which is characterized by good BBB permeability in rodents and specific linking as seen in autoradiography in rodents and human hippocampal sections. From these studies, it appears that ^18^F-FEBMP and ^11^C-Ac5216 are promising tracers thanks to their low non-specific cerebral vessel uptake. On the other hand, [^11^C]CPPC seems to be a better choice since it has a specific brain uptake and is not influenced by the polymorphism of the TSPO gene. The same consideration could be performed for the novel [^11^C]NCGG401, even if further studies on humans are required.

Regarding neuroinflammation, we have to cite progressive supranuclear palsy (PSP). It is a neurodegenerative disorder caused by the aggregation of misfolded hyperphosphorylated four-repeat tau proteins that present as oligomers and, after they tangle, enter the basal ganglia, diencephalon, and brainstem [37,38]. Neuroinflammation is the other side of the coin that often colocalizes with tau deposition in PSP [39,40,41]. Clinically, PSP is characterized by a plethora of symptoms, named Richardson’s syndrome, which includes vertical supranuclear gaze palsy, cognitive decline, akinetic rigidity, and balance disorders [42,43]. The possible correlation between tau aggregation and microglia activation could be an interesting aspect to better characterize in imaging, which can be a significant help in the definition of diagnosis. Malpetti et al., valued this correlation using [^18^F]AV-1451 and [^11^C]PK11195 for microglial activation [44]. They hypothesized that neuroinflammation and tau aggregation could be present in the same areas and could correspond to the severity of the syndromic clinical presentation. Non-displaceable binding potential (*BP*_ND_) for [^11^C]PK11195 correlated with [^18^F]AV-1451 binding in the brain, with the highest levels in the brainstem, cerebellum, thalamus, and occipital and cingulate cortex for [^11^C]PK11195. Relevant uptake of [^18^F]AV-1451 was displayed in the basal ganglia, midbrain, and thalamus. A positive correlation between clinical severity, subcortical tau pathology, and neuroinflammation was revealed. [^18^F]AV-1451 binds to aggregated tau in AD and has low specificity since it cannot differentiate TDP43 pathologies from tau. Despite this premise, in this study, the author focused on the relevant clinical and anatomopathological correlations in PSP. However, TDP43 pathologies are rare in PSP. Moreover, the pattern of tracer distribution is typical in PSP and AD, so a differential image-based diagnosis could be performed.

The hypothesis that neurodegeneration could be due not only to protein misfolding but also to neuroinflammation played a role in the efforts to develop new tracers able to detect different targets, the so-called multi-target-directed ligands (MTDLs). An example is the paper of Zhu et al., in which a series of novel O-alkyl ferulamide derivatives were synthesized. Ferulic acid (FA) is a polyphenol particularly represented in maize bran that seems capable of potentiating the inhibition of monoamine oxidase B (MAO-B) if benzyl derivatives and alkyl fragments were introduced into its skeleton [45]. Monoamine oxidases A (MAO-A) and B are responsible for the degradation of neurotransmitters such as dopamine, serotonin, adrenaline, and noradrenaline. MAO-A and B have increased activity in AD and are considered involved in neuroinflammation through the production of hydroxyl radicals, which cause the generation of Aβ plaques [46]. Zhu and colleagues produced 5a, 5d, 5e, 5f, and 5h compounds and found good antioxidant and MAO-B inhibitory activity and relevant blood–brain barrier (BBB) penetration. After the radiolabeling with ^11^C, compound 5f demonstrated a good clearance kinetic property and a good BBB penetration, while the blocking studies with Rivastigmine showed a specific binding of the tracer (as seen in Figure 2).

### 4.2. PET Tracers for Specific Tau Filaments

Tau protein has a pivotal role in the pathology of different diseases: Alzheimer’s disease (AD), chronic traumatic encephalopathy (CTE), progressive supranuclear palsy (PSP), globular glial tauopathy (GGT), corticobasal degeneration (CBD), and Pick’s disease (PiD). Tau isoforms in humans are six; three have three microtubule-binding repeats, and the other has four binding repeats. All isoforms are present in AD and CTE, while 4R is frequently found in PSP, GGT, and CBD, while PiD is characterized by the 3R isoform [47]. Different tracers have been developed to identify specific ligands of the tau protein. Shi et al., studied ^18^F-APN-1607, also known as ^18^F-PM-PBB3 (a propanol modification of PBB3), to establish the specific binding region in paired helical and straight filaments (PHFs and SFs) of tau in AD [48]. Through post-mortem cryo-microscopy (cryo-EM) conducted on human brains, it was detected that there were two sites in the β-helix of PHFs and SFs and another binding site in the SFs’ C-shaped cavity.

On the same philosophy of research, there is the work of Schröter et al., who performed an in vivo PET scan in 4R tauopathies on two patients with corticobasal syndrome (CBS), seven with PSP, and two patients with AD using ^11^C-pyridinyl-butadienyl-benzothiazole 3 (^11^C-PBB3) as a tracer [49]. They found a specific and different spatial distribution of the tracer: the uptake was more relevant in the dorsolateral frontal and motor cortex of CBS subjects; higher uptake was evident also in frontotemporal areas of AD patients, at the opposite of PSP ones, which presented a slight uptake in the midbrain, more relevant compared with AD patients.

Interestingly, in another study, it was observed that the novel [^18^F]PI-2620 had different binding properties between AD, CBD, and PSP. It was assessed through molecular docking, molecular dynamics simulation, metadynamics, and Brownian dynamics [50]. The best binding site in AD–tau filaments has been found in the C-shaped groove of the filament core, while in CBD and PSP, it is in the inner filament core. Sites of binding on the outer surface had high binding free energy but were short-lived. In CBD and PSP, the kinetics were limited, while AD binding was characterized by favorable kinetics.

Another tubular PET study is that of Kumar et al. [51]. They tested ^11^C-MPC-6827 in transgenic mice bearing tau pathology (rTg4510) and amyotrophic lateral sclerosis pathology (SOD1*G93A). This population underwent a micro-PET scan, and a lower binding was highlighted compared to wild-type littermates.

^11^C-MPC-6827 could be useful in AD imaging, amyotrophic lateral sclerosis, and other tauopathies, while ^18^F-PM-PBB3, [^18^F]PI-2620, and ^11^C-PBB3 could be more advantageous for differential diagnosis between tauopathies.

### 4.3. Protofibril Tracers

A novel philosophy of research focused on nonfibrillar Aβ oligomers and protofibrils since they appeared to be better biomarkers of clinical severity compared to Aβ plaques [52]. Meier and colleagues based the study on the PET antibody approach [53]. A comparison between ^11^C-Pittsburgh compound B (^11^C-PiB) and the protofibril tracer ^124^I-RmAb158-scFv158 (Aβ protofibril selective antibody mAb158) to evaluate the effects of the β-site APP cleaving enzyme (BACE)1 inhibitor NB-360 has been conducted. BACE-1 is a 501-amino acid type 1 transmembrane aspartic protease that is responsible for APP cleavage as the first step in the formation of the pathogenic amyloid-β peptides [54,55]. In the Meier et al., experiment, the BACE-1 inhibitor NB-360 was administered to 16 month-old transgenic ArcSwe mice for 2 months, while a different group was kept as a control. A PET scan with ^11^C-PiB was conducted to assess the amount of Aβ plaque. The scan was repeated with the bispecific radioligand ^124^I-RmAb158-scFv8D3 that links to non-fibrillar Aβ aggregates. The same procedure was applied to the 8 month-old AppNL-G-F mouse model treated with NB-360. The results demonstrated a low uptake of ^124^I-RmAb158-scFv8D3 in the thalamus and hippocampus of NB-360-treated ArcSwe mice, lower than the untreated counterparts, and in the cerebellum of NB-360-treated AppNL-G-F mice. Treated mice have no lower ^11^C-PiB uptake than untreated mice. This study highlights how ^124^I-RmAb158-scFv8D3 is a more sensitive tracer, maybe thanks to the conjugation with an antibody that makes the binding more selective and specific.

Another exciting work is the study of Zeydan et al., which started with the fact that the most commonly used ^11^C-PiB correlates negatively with white matter hypersensitization (WMH) seen in magnetic resonance [56]. WMH increases with aging but is characterized by lower tracer uptake, suggesting independence between WMH and ^11^C-PiB uptake. The short half-life of ^11^C led the researchers to the ^18^F-flutemetamol PET ligand for white matter integrity. Sixty-one cognitively unimpaired (CU) older and younger adults underwent MRI and PET scans with the two tracers. They analyzed the SUVr in WMH and normal-appearing white matter (NAWM) and found no differences between them, even if the uptake was age-correlated. These results confirmed the similar uptake pattern between the two tracers.

### 4.4. Comparative Studies between Amyloid Tracers

The FDA-approved visual rating guidelines used to determine whether a scan is positive or negative vary greatly in terms of intensity scaling, region definitions, color scales, and spatial and signal thresholds. The reading results of amyloid PET performed on the same subject using various ligands can differ. A comparative study by Cho et al., evaluated the differences between ^18^F-florbetaben (FBB) and ^18^F-flutemetamol (FMM) amyloid PET. The study explored the discrepancy in detecting amyloid positive between these two tracers through the ocular evaluation, the cortical-to-reference region standardized uptake value ratio (SUVR), and the direct comparison of FBB-FMM centiloid (dcCL) [57]. A total of 107 subjects underwent PET scans; of these, 64.4  ±  17.2 years was the average age, 56.1% were females, and SUVr was calculated taking the cerebellum as the reference area. The visual estimation has a concordance rate between FBB and FMM of 94.4%, while the false-positive rate was higher in FMM (9.1%) than in FBB (1.8%) when the two ligands were analyzed using SUVR cut-off categorization, although these results were not statistically significant (*p* = 0.13). The research documented that FBB and FMM had an excellent agreement in quantitative and qualitative analysis, and it is possible to combine amyloid PET data with the use of various ligands from multiple centers. The results are of high value since the number of participants is high and are the first step for standardization between centers.

The same group continued to compare the uptake in several regions in more detail. They recruited 107 patients, classified as follows: 20 were young healthy controls, 27 were old controls, 27 subjects with mild cognitive impairment (MCI), 29 subjects with AD, and four subjects with subcortical vascular dementia (SVaD) [58]. The results indicated a correlation between the two tracers’ cortical uptake when the cerebellum was considered the reference region. Only the striatal uptake was more relevant in FMM than in FBB, suggesting only for striatal studies that FMM has better diagnostic accuracy.

An important aspect is related to the binding properties of [^18^F]flortaucipir: its retention has been found in vivo to correlate strongly with post-mortem tau pathology in MAPT R406W mutant gene carriers and with Alzheimer’s disease-related tau pathology [57,59]. Furthermore, [^18^F]flortaucipir distinguishes AD from other neurodegenerative disorders based on tracer retention in the temporal cortex [59]. However, [^18^F]flortaucipir is retained significantly in the basal ganglia, thalamus, and choroid plexus, where no paired helical filaments (PHFs) of tau pathology are expected. Tau pathology associated with AD is characterized by PHFs, a mixture of three- and four-repeat tau isoforms. When compared to [^18^F]flortaucipir, novel tau PET tracers such as [^18^F]RO948, [^18^F]PI-2620, [^18^F]GTP1, and [^18^F]MK-6240 have slightly different binding properties [59]. Therefore, [^18^F]flortaucipir and [^18^F]RO948 were compared in the study of Smith et al., and in 18 participants with AD, three of them had amyloid-β-positive amnestic mild cognitive impairment and four controls [59]. The neocortical retention was similar in both tracers. Differences could be detected in the entorhinal cortex, where [^18^F]RO948 uptake was higher, and in the basal ganglia, thalamus, and choroid plexus, where [^18^F]flortaucipir retention was lower. Non-target uptake was evident in both tracers. In some cases, important accumulations of [^18^F]RO948 were observed in the skull and meninges, but they remained unchanged in revaluation after 1 year. However, the off-target intracerebral uptake was lower in this compound than in [^18^F]flortaucipir. Despite the restricted population number, this study is distinguished for the 1-year follow-up, which adds an additional diagnostic value.

FMM seems to be a more accurate tracer for striatal studies, even if no differences in visual estimation have been found between FMM and FBB. No differences in cortical uptake have been seen between [^18^F]flortaucipir and [^18^F]RO948, even if the last one has lower unspecific intracerebral uptake. These results suggest the high diagnostic accuracy of these tracers, which can all be roughly considered valid candidates for amyloid PET imaging.

### 4.5. Distribution Studies in Patients with Tauopathies

Another in vivo evaluation was conducted by Mormino et al., which underwent an ^18^F-PI-2620 tau PET scan on six healthy subjects, 11 beta-amyloid positive patients with cognitive impairment (with a clinical diagnosis of AD or mild cognitive impairment), and two humans with semantic variant primary progressive aphasia (svPPA) [60]. In AD patients, the medial temporal lobe was particularly affected, while the cortical region in cognitive impairment subjects demonstrated more variability (only one had higher uptake in the posterior cingulate and lateral parietal cortex). In one of the two subjects with svPPA, a focal uptake was revealed in the anterior temporal pole. In atypical AD, an important signal was detected in the posterior association cortices. AD clinical severity is demonstrated to be correlated to [^18^F]-AV1451, and in particular, increased frontal, medial temporal, and occipital cortices uptake are associated with psychosis and fast cognitive and functional deterioration [61].

An important step forward in the differential diagnosis between corticobasal syndrome (CBS) subtypes is the feasibility of using [^18^F]PI-2620, which stands out for its less off-target binding to monoamine oxidases, relevant affinity to three repeats/four repeats (3R/4R) tau in AD, and affinity also for 4R tau in PSP [62]. CBS is a rare adult-onset disease characterized in around 50% of cases by 4R tau aggregation, in 25% of mixed 3R/4R tau isoforms, and in the remaining 25% by non-tauopathies. Therefore, 45 patients with corticobasal syndrome and 14 age-matched healthy subjects underwent [^18^F]PI-2620 PET scans. [^18^F]Flutemetamol or [^18^F]Florbetaben PET imaging was previously conducted in some controls to detect Aβ plaque. High accumulation of [^18^F]PI-2620 was seen in the dorsolateral prefrontal cortex and basal ganglia in both amyloid-positive and negative subjects, but was superior in Aβ positive ones compared to healthy controls. From the clinical perspective, no relevant correlation was found between cognition and tracer uptake, even if this could be affected by the discrepancy between symptom onset and clinical assessment of cognitive and motor dysfunction. A limited tracer sensitivity has been assessed for a diagnosis suggestive of PSP–CBS, and the restricted population number and the lack of autopsy analysis required further investigations, even if, given the rarity of the disease, this study is an excellent initial evaluation.

The necessity of the development of a specific 4R tau PET tracer remains. Tsai et al. [63] examined the tau tracer ^18^F-flortaucipir in 11 patients with nonfluent variant primary progressive aphasia (nfvPPA), 10 with corticobasal syndrome (CBS), 10 bvFTD subjects, two with svPPA, and six with FTD-associated pathogenic genetic mutations, microtubule-associated protein tau (MAPT), five with chromosome nine open reading frame 72 (C9ORF72) association, and one associated with progranulin (GRN) mutation. Patients with nfvPPA had higher ^18^F-flortacupir uptake in the left inferior frontal gyrus than the right, while subjects with CBS exhibited increased frontal white matter binding, and a subset of amyloid-positive patients also had higher cortical grey matter uptake. In sporadic bvFTD, frontotemporal binding was shown to be enhanced in five out of ten patients. There were no detected differences between CBS and bvFTD. A post-mortem examination on a patient with C9ORF72, TDP-43-type B pathology, and neurofibrillary tangles in the middle frontal and inferior temporal gyrus revealed mild ^18^F-flortaucipir uptake. A patient with sporadic bvFTD had higher uptake in the inferior temporal and hippocampus, which corresponded to areas of severe argyrophilic grain disease pathology. These results confirmed, as seen in other studies, the restricted sensibility and specificity of this tracer.

Wren et al., conducted a microscopic neuropathological analysis of human brain tissue using nuclear imaging [64]. They studied [^18^F]flortaucipir, the derivative [^18^F]T808, and the fluorescent analog T726, which have all in common the structure (class of carbazoles), and [^18^F]THK-5105, [^18^F]THK-5117, and [^18^F]THK-5351 that have a 2-arylquinoline core. Alzheimer’s disease cases and one case with frontotemporal dementia and parkinsonism connected to chromosome 17 expressing an R406W MAPT mutation showed carbazole and 2-arylquinoline binding. PET imaging with [^18^F]THK-5117 confirmed that there is considerable inter- and intra-case variability in tracer binding in end-stage Alzheimer’s disease cases. It was discovered by microscopic examination of the diseased inclusions that the fluorescent tracers bind premature tau aggregation preferentially. THK-5117 also showed neuritic tau binding, whereas T726 binding was restricted to neuronal tau. Neither tracer showed tau linking in the pre-symptomatic stage. These results affirmed the limitations of first-generation tau tracers, whose binding is not correlated to the severity of tau deposition and has low sensitivity in the early stages.

### 4.6. Novel Radiotracers to Image Cognitive Decline

As said before, AD has in common with tauopathies the presence of intraneuronal tau inclusions [10]. Therefore, although AD is considered a secondary tauopathy, it could be interesting to cite other thrilling advances in neuroimaging related to cognitive decline in Alzheimer’s disease.

It is known that the loss of α4β2 nicotinic acetylcholine receptors (nAChRs), the most common subtype in humans, can be seen in AD with the subsequent alteration of the cholinergic system. The loss of the α4 subunit binding site correlates with dementia’s severity and with the β-amyloidosis process [65,66,67]. The first estimation of the nAChRs’ distribution in vivo was realized through [^11^C]nicotine. Since [^11^C]nicotine retention in the brain is influenced by blood flow, blood–brain barrier transport, and unspecific binding, a precise correlation of changes in [^11^C]nicotine accumulation with nAChR availability was impossible to establish. This problem was partially solved by 3-pyridyl ether derivatives because their slow kinetics limit their use in the clinical routine. New PET ligands with more favorable properties have been created, such as homoepibatidine, epibatidine, or 3-pyridylether derivatives. An example is (+)-[^18^F]Flubatine, characterized by favorable data for dosimetry, picture quality, and kinetics, as well as a low amount of metabolites. Moreover, it has a higher binding affinity and a similar metabolism but slower kinetics compared with its enantiomer (−)-[^18^F]Flubatine. This led to the consideration that applying a reference region to the analysis of (+)-[^18^F]Flubatine cerebral distribution is not necessary [64]. Tiepolt and colleagues evaluated eleven healthy subjects and nine with mild AD through [^11^C]PiB PET/MRI and (+)-[^18^F]Flubatine PET scans. Full kinetic modeling for (+)-[^18^F]Flubatine was realized by a 1-tissue compartment model. In the patients with mild cognitive impairment, a decreased availability of α4β2 nAChRs was seen in the bilateral mesial temporal cortex. Interesting correlations between white matter integrity and α4β2 nAChR availability were found between white matter amyloid PET uptake and (+)-[^18^F]Flubatine binding. This study assessed the feasibility of (+)-[^18^F]Flubatine use in humans for its stability and safety [65].

The connection between neurological impairment and genetic syndromes has long been investigated. A postmortem study was conducted by Lemoine et al., on subjects with Down syndrome and AD (DS–AD) [68]. DS increases the risk of developing AD due to the triplication of chromosome 21, where the amyloid precursor gene (APP) is located [69].

The first-generation Tau tracer THK5117 has a distinct laminar binding pattern in some cortical areas in patients with AD, even if tau binding cannot be detected in people with DS before the age of 30; it is unknown whether this is due to PET sensitivity limitations or reduced binding to certain forms of pathological Tau. Moreover, tau tracers of the second generation, including MK6240, have been produced [68]. Therefore, the authors used autoradiography to assess the preferential binding of two tau tracers (^3^H-MK6240 and ^3^H-THK5117) and one amyloid (^3^H-PIB) ligand in the medial frontal gyrus (MFG) and hippocampus (HIPP) between DA-AD and sporadic AD. Adjacent brain sections were immunoreacted for phosphor-Tau (AT8) and amyloid using Amylo-Glo staining. ^3^H-MK6240 linked AT8 immunostaining sections, but in a lower grade compared to ^3^H-THK5117. High binding density was evident in the HIPP and MFG of DS patients for the three tracers. In adults with DS, a higher binding density of THK was evident compared to AD. No correlation was observed between ^3^H-PIB and Amylo-Glo straining in adults with the diagnosis of DS, which demonstrated the existence of additional sites for the binding of amyloid tracer. MK6240 has also been radiolabeled with ^18^F, and the resulting radioligand, ^18^F-MK-6240, was investigated in humans by Betthauser and colleagues [70]. High uptake was seen in the pons and the inferior cerebellum, in concordance with neuropathological neurofibrillary tau staging, with minimal off-target uptake shown in the ethmoid sinus, clivus, meninges, and substantia nigra. In Figure 3, the specific uptake and off-target uptake of ^18^F-MK-6240 are illustrated in PiB-positive AD and MCI subjects and PiB-negative patients.

A new target for AD is the sigma 1 receptor (S1R), a protein of the endoplasmic reticulum expressed in the CNS. S1R is involved in calcium signaling related to cellular survival and in cholinergic, noradrenergic, and dopaminergic modulation. Its density is low in AD and other neuropathies, so it represents a valid biomarker of neuronal function. To better understand its potential use in neurologic diagnosis, a ligand of S1R, ^18^F-IAM6067, was studied in mice and human brains [71]. The tracer distribution was from high to low as follows: pons-raphe, thalamus mediodorsal, substantia nigra, hypothalamus, cerebellum, cortical areas, and striatum. No differences were detected between patients with AD and healthy subjects. This result suggested that this compound may not be the ideal tracer in humans and that there are no advantages over the well-known (S)-^18^F-fluspidine for imaging S1R in humans.

The cytoplasmic serine/threonine protein kinase Glycogen synthase kinase-3β (GSK-3β) was studied by Zhong and colleagues since it is involved in different biological processes, from glycogen metabolism, cell signaling and transport, cellular apoptosis, and neurogenesis, to the pathogenesis of AD, and it is well represented in human cortical regions, the locus coeruleus, the hippocampus, and the amygdala, while low concentrations are in the striatum. Additionally, the existing tracers for this target, such as [^11^C]PyrATP-1, [^11^C]CMP, [^11^C]AR-A014418, [^11^C]A1070722, and the series of [^11^C] labeled oxadiazole analogs, have poor brain accumulation, or in the cases of [^11^C]SB-216763, [^11^C]PF-367, and [^11^C]oxazole-4-carboxamide analogs, they have homogeneous brain uptake. Therefore, the researchers synthesized the new compound [^18^F]10a–d, which is a nicotinamide derivative. It showed high affinity, good blood–brain barrier penetration in rodents, and specificity to the target areas rich in GSK-3β, such as the amygdala, cerebellum, and hippocampus [72].

An innovative therapeutic drug candidate for cognitive impairment in AD is the agonist of α7-Nicotinic acetylcholine receptor (α7-nAChR), which is a ligand-gated ion channel responsible for acetylcholine (ACh) binding. Thus, this receptor seems involved in cognitive decline, and lower expression has been identified in AD subjects [73,74]. Wang et al., proposed a novel class of 1,4-diazobicylco[3.2.2]nonane derivatives as α7-nAChR ligands for PET scans [75]. They synthesized and radiolabeled compound 15 with ^18^F, which showed at 15 min post-injection in rats a high brain uptake and specific labeling confirmed in blocking studies. The pioneering method of Wang et al., consists of using transferrin receptor (TfR)-mediated transcytosis [76,77,78]. OX265 and OX2676, two variants of mouse anti-rat Tfr (rTfR), were synthesized and linked to bapineuzumab (Bapi), an anti-Aβ antibody F(ab’)2 fragment, generating OX265-F(ab’)2-Bapi and OX2676-F(ab’)2-Bapi [79]. The two compounds were radiolabeled with ^124^I and ^125^I. The results indicated a higher brain uptake for [^125^I]I-OX265-F(ab’)2-Bapi after 4 h from the injection in TgF344-AD rats bearing human APP with the Swedish mutation (AβPP KM670/671NL) and human PSEN1 with exon nine deletion (PS1-ΔE9). [^124^I]I-OX265-F(ab’)2-Bapi uptake was more evident in TgF344-AD rats than in controls, indicating a specific binding and assessing the utility of this approach. This tracer seems to be promising compared to the already known [^125^I]1, [^11^C]CHIBA-1001, [^18^F]NS10743, and [^18^F]ASEM, which show low brain uptake or low specific binding.

### 4.7. Clearance and Biodistribution Studies of Two Amyloid and Tau Radiopharmaceuticals

In the framework of AD diagnosis, the well-known Aβ plaque [^11^C]PiB PET tracer has the pitfall that it saturates early in the disease’s development and is not able to identify soluble or diffuse Aβ pathology [80]. The advent of molecular imaging, which is accompanied by antibodies, permitted the exploitation of the brain via receptor-mediated transcytosis. However, the long circulation period of these antibodies generates a problem of radiation exposure [81], which was considered in the challenging work of Schlein and colleagues, whose aim was to ameliorate the clearance of similar compounds and enhance imaging contrast [82]. They used RmAb158-scFv8D3, a bispecific Aβ targeting antibody, and RmAb158, a monospecific antibody, functionalized with trans-cyclooctene or -d-mannopyranosylphenyl isothiocyanate (mannose) (TCO), both characterized by liver clearance. In transgenic AD mice, while RmAb158 circulation was limited thanks to the greater cited functionalization, limited effects were observed for RmAb158-8D3. The injection of a tetrazine-functionalized clearing agent boosts the contrast of TCO-[^125^I]I-RmAb158 in AD mice a day post-injection.

The first toxicity study was conducted on rats, and dosimetric and biodistribution evaluations of [^18^F]MK-6240 were conducted on humans [83]. The 7-day repeat-dose intravenous (IV) toxicity study in rats did not show any adverse events at a dose ≥ 333 μg/kg/day, which corresponds to 1000-fold over the maximum clinical dose of 0.333 μg/kg. No hemolysis was observed in human or rat blood. Human whole-body imaging showed the highest initial uptake in the liver and higher retention in the gallbladder (the organ with the major absorbed dose), intestines, and urinary bladder due to hepatobiliary and renal clearance of the compound, with an average effective dose equal to 29.4 ± 0.6 μSv/MBq.

### 4.8. Studies concerning Important Genes and Enzymes Involved in Neurodegeneration

The microtubule-associated protein tau gene (also known as MAPT), in particular the variant rs242557, has a close correlation with tauopathies and dementia since it encodes tau protein for the stability of microtubules and transduction of signals [84]. In particular, the H1 haplotype is associated with AD, PSP, and corticobasal degeneration (CBD) [85]. The variant rs242557 single nucleotide polymorphism (SNP) is located in a regulatory region of MAPT and interferes with its expression [86,87]. Different studies have explored the correlation between this SNP and plasma and cerebrospinal fluid tau levels [88,89], so Shen et al., evaluated the possible correlation with imaging in 90 non-demented elders through PET with ^18^F-AV-1451 [90]. Based on standardized uptake value ratios (SUVRs), they found an increased uptake in the hippocampus of non-demented subjects with minor allele A, exactly as Aβ-positive subjects and APOE ε4 non-carriers. However, the small population and the single tracer study represent limits that reduce testing efficiency. Besides that, no anatomopathological analysis has been performed.

Jones et al., have yet to explore the possibility of examining ^18^F-AV-1451 uptake among microtubule-associated protein tau (MAPT) mutation carriers [91]. Ten symptomatic and three asymptomatic MAPT mutation carriers with clinically normal (CN) subjects and AD dementia were included. Eight participants had MAPT mutations on exon 10 and tended to form 4R tau pathology, and five had mutations outside exon 10 and formed mixed 3R/4R tau pathology. Higher uptake was evident in AD and minimal in MAPT mutation carriers with a mutation on exon 10, while higher uptake, similar to the AD pattern, was observed in patients with MAPT mutations outside exon 10.

Also, in the multicenter study of Matsuda and colleagues, the substantial clinical impact of ^18^F-florbetapir in AD and non-AD diagnosis was established [92], while in the work of Lesman-Segev et al., it was demonstrated the higher sensitivity of ^11^C-PiB PET compared to ^18^F-fluorodeoxyglucose PET for intermediate-high AD neuropathological change but similar specificity [93]. A similar study was that of Gordon and colleagues. They studied the distribution of ^11^C PiB and ^18^F-AV-45 (florbetapir) in patients with dominantly inherited Alzheimer’s disease [94]. The pattern of tau ligand distribution was the same as in sporadic Alzheimer’s disease, although the inherited patients had more cortical involvement and higher levels of binding despite having a similar cognitive impairment. In addition to the temporal lobe, significant tau uptake was evident in the precuneus and lateral parietal regions. Symptomatic mutation carriers also showed enhanced tau PET binding in the basal ganglia. An increased binding was highlighted in amyloid in both the asymptomatic and symptomatic groups.

In the report of Paul et al., the pharmacokinetics of ^18^F-LSN3316612 were studied [95]. This PET tracer has high affinity and selectivity for O-GlcNAcase (OGA), which is involved in tau phosphorylation. PET scans were conducted on monkeys; in the brain, the uptake was high and characterized by a slow washout. The highest accumulation was detected in the amygdala, striatum, and hippocampus. Pretreatment with thiamet-G (an OGA inhibitor) or the non-radioactive LSN3316612 significantly decreased brain uptake. The whole-body evaluation showed high uptake in the kidney, spleen, liver, and testes. PET imaging was also performed in Oga^∆Br^, a mouse brain-specific knockout of Oga that had low brain activity compared to normal mice.

In Table 1, the above-mentioned studies are categorized by tracer, other possible imaging modalities included in the study, and population.

## 5. Conclusions

The research of novel radiotracers for the diagnosis of tauopathies is challenging. Non-target binding or rapid saturation during the clinical evolution of the disease is the principal pitfall of the current ligands. The necessity of more specific imaging tools led to the search for new targets with the help of genetics and nanomedicine. Nowadays, the necessity to experience novel ligands in humans requires ulterior efforts, burdened by the fact that, apart from some diseases such as Alzheimer’s disease, many tauopathies are rare and it is not always possible to perform post-mortem studies.

## Figures and Tables

**Figure 1 diagnostics-13-01682-f001:**
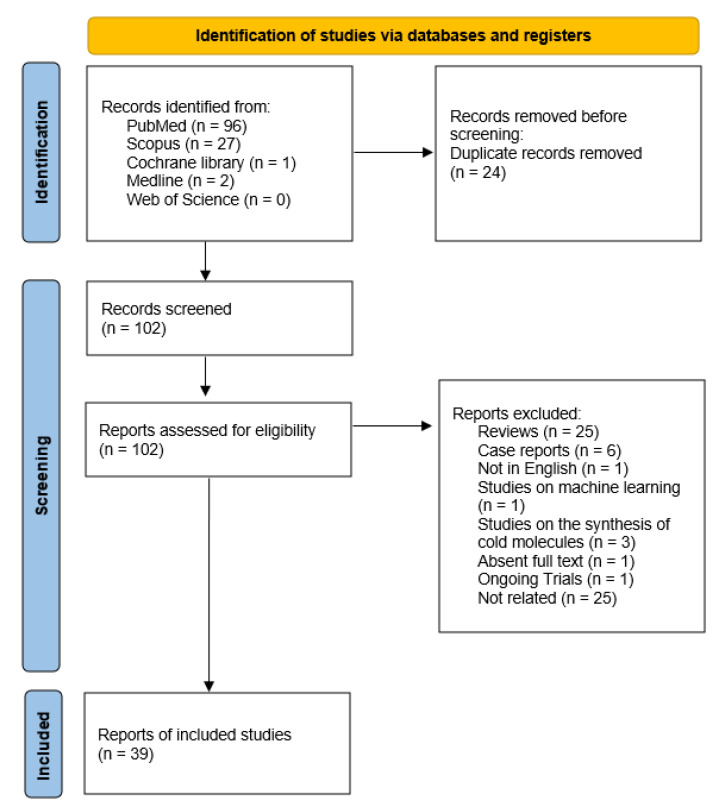
PRISMA 2020 flow diagram.

**Figure 2 diagnostics-13-01682-f002:**
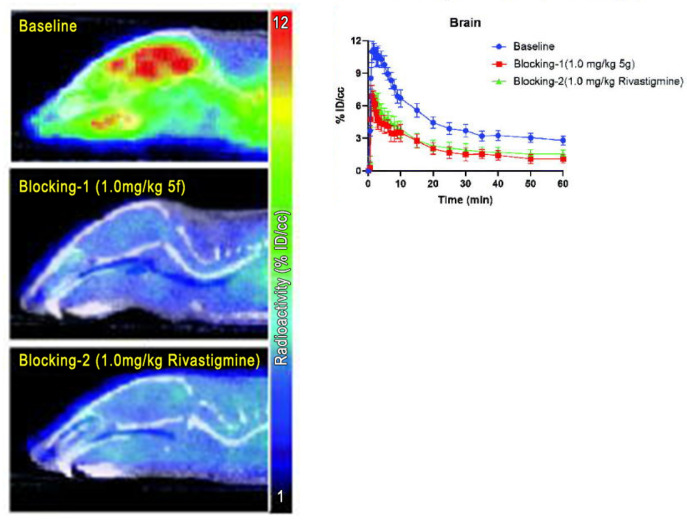
From Zhu et al., PET/CT studies of brain uptake of [^11^C]5f in mice at baseline, and after unlabeled [^11^C]5f and Rivatigmine pre-treatment and representation of percentage of tracer’s injected dose per unit volume in function of time in mice at baseline (blue line) and after the administration of unlabeled [^11^C]5f (red line) and Rivatigmine (green line) [44].

**Figure 3 diagnostics-13-01682-f003:**
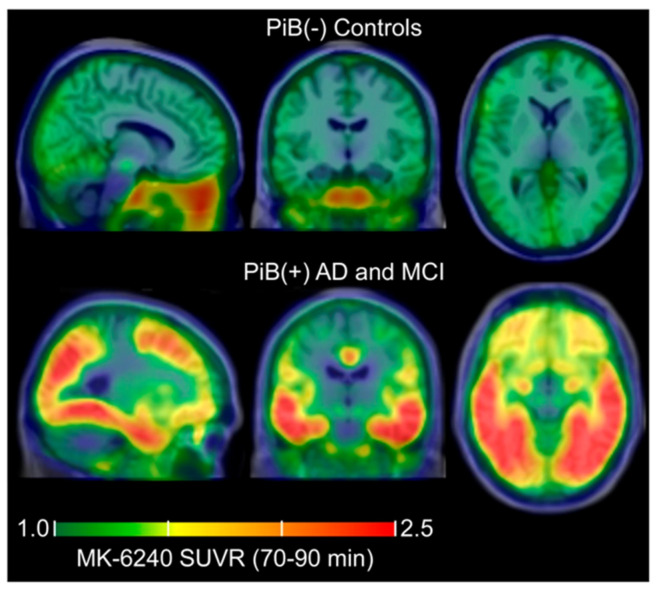
From Betthauser and colleagues, ^1^⁸F-MK-6240 uptake in PiB-negative, PiB-positive AD, and PiB-positive MCI patients. Off-target uptake is seen in the ethmoid sinus, clivus, meninges, and substantia nigra [67].

**Table 1 diagnostics-13-01682-t001:** Classification of the included studies. Here are reported the tracer object of research, the other imaging modalities considered in the paper, and the type of population chosen for the experiment.

Study	Studied Tracer	Other Imaging Modalities Considered	Study Conducted on
Malpetti et al. [14]	^18^F-AV-1451	MRI	Humans
	^11^C-PK11195		
Rauchmann et al. [16]	[^18^F]GE-180	MRI	Humans
Fairley et al. [17]	^11^C-PBB3 ^18^F-FEBMP	MRI	Mice
Bevan-Jones et al. [21]	^11^C-PK-11195 ^18^F-AV-1451	/	Humans
Bevan-Jones et al. [22]	^18^F-AV-1451	/	Humans
Ji et al. [23]	^18^F-FEBMP ^11^C-PK11195 ^11^C-PBR28 ^11^C-Ac5216 ^18^F-FEDAA1106	/	Mice
Horti et al. [25]	[^11^C]CPPC	/	Murine model
Ogata et al. [36]	[^11^C]NCGG401	/	Rodents, Humans
Malpetti et al. [44]	[^11^C]PK11195, [^18^F]AV-1451	/	Humans
Zhu et al. [45]	[^11^C]5f	/	Mice
Shi et al. [48]	^18^F-APN-1607	/	Humans
Schröter et al. [49]	^11^C-PBB3	/	Humans
Künze et al. [50]	[^18^F]PI-2620	/	Simulation Workflow
Kumar et al. [51]	^11^C-MPC-6827	/	Mice
Meier et al. [53]	^124^I-RmAb158-scFv158, ^11^C-PiB	/	Mice
Zeydan et al. [56]	^11^C-PiB ^18^F-flutemetamol	MRI	Humans
Cho et al. [57]	^18^F-florbetaben ^18^F-flutemetamol	/	Humans
Cho et al. [58]	^18^F-florbetaben ^18^F-flutemetamol	/	Humans
Smith et al. [59]	[^18^F]flortaucipir [^18^F]RO948	/	Humans
Mormino et al. [60]	^18^F-PI-2620	/	Humans
Gomar et al. [61]	[^18^F]-AV1451	/	Humans
Palleis et al. [62]	[^18^F]PI-2620 [^18^F]Flutemetamol [^18^F]Florbetaben	/	Humans
Tsai et al. [63]	^18^F-flortaucipir	/	Humans
Wren et al. [64]	[^18^F]flortaucipir [^18^F]T808	/	Humans
Tiepold et al. [65]	(+)-[^18^F]Flubatine	/	Humans
Lemoine et al. [68]	^3^H-MK6240 ^3^H-THK5117 ^3^H-PIB	/	Humans
Betthauser et al. [70]	^18^F-MK-6240	/	Humans
Lepelletier et al. [71]	^18^F-IAM6067	/	Mice Humans
Zhong et al. [72]	[^18^F]10a-d	/	Rodents
Wang et al. [75]	[^18^F]15	/	Rats
Bonvicini et al. [79]	[^125^I]I-OX26_5_-F(ab’)_2_-Bapi [^124^I]I-OX26_5_-F(ab’)_2_-Bapi	/	Rats
Schlein et al. [82]	TCO-[^125^I]I-RmAb158	/	Mice
Koole et al. [83]	[^18^F]MK-6240	/	Rats Humans
Shen et al. [90]	^18^F-AV-1451	/	Humans
Jones et al. [91]	^18^F-AV-1451	/	Humans
Matsuda et al. [92]	^18^F-florbetapir	/	Humans
Lesman-Segev et al. [93]	^11^C-PiB ^18^F-fluorodeoxyglucose	/	Humans
Gordon et al. [94]	^11^C PiB ^18^F-AV-45	/	Humans
Paul et al. [95]	^18^F-LSN3316612	/	Monkeys Mice

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
