# Peer review of "Imaging of Tauopathies with PET Ligands: State of the Art and Future Outlook"

_diagnostics, 2023, doi:10.3390/diagnostics13101682_

Round 1
Reviewer 1 Report
This manuscript submited by Dr. Frantellizzi conducted a systematic search with keywords “PET ligands” AND “tauopathies” in several database and analyzed the recent development on the PET ligand targeting tauopathies. A major revision is recommended before the acceptance for publication.
1. This manuscript laid out the results from the chosen literature, please consider adding some discussion on the advantages and disadvantages of each study, which would be more consistent with the title “state of art and future outlook”.
2. In general, this manuscript is written redundantly. Some non-relevant/unimportant details of the discussed studies were included in the manuscript. There are also repititions on the explanation of the same tracer, for example 18F-AV-1451, as well as repititions on the details of the same group of subjects, for example ref 55 and 56 discussed different aspects of the same study group.
3. Please add a brief discussion on how the comparative study on amyloid tracers can benefit the understanding of Tau pathology in section 3.4
4. Please specify the target type/category of novel ligands in the description of the thematic area #6 as “studies on novel ligands” is rather broad as a catagory. A better rearrangement of the contents in section 3.6 is highly recommended, for example the studies on MK-6240 (ref 65, 67 and 80) should be discussed together. Please also add a brief discussion on the correlation of the various protein (a4b4, sigma-1, a7 tracers etc.) function and tau pathology. Currently, this section is more like a discussion on the PET ligands for AD, although AD is considered as a secondary tauopathies.
The English language in this manuscript needs to be polished more. Some sentences are not easy to understand, and grammar mistakes and typos are also seen in several places.
Author Response
First of all we would like to thank you for the attention shown and the precious suggestions made, which we have welcomed. We have been glad to put them in place, conscious of the fact that your advice and suggestions will certainly improve the quality and comprehensibility of our work. All recommended corrections have been made and underlined in the manuscript, as detailed below.
- The lacking discussion between the cited studies has been added throughout the paper.
- Repetitions has been deleted. The introduction to ref 56 has been modified as follows: “The same comparison has been done between the two tracers” since it is in the same group as ref 55
- A brief discussion has been added in section 3.4 as follows: “FMM sems to be a more accurate tracer for striatal studies even if no differences in visual estimation have been found between FMM and FBB. No differences in cortical uptake have been seen between [18F]flortaucipir and [18F]RO948 even if the last one has lower unspecific intracranial uptake. These results suggest the high diagnostic accuracy of these tracers that can be roughly considered all valid candidates for amyloid PET imaging.“
- Paragraph 3.6 has been renamed “Nicotinic radioligands, novel tau, and amyloid radiopharmaceuticals, Glycogen synthase kinase-3β and Sigma 1 receptor tracers” and it has better explained the correlation between ref 65 and 67 while ref 80 has been included in the subparagraph “3.6.1 Clearance and biodistribution studies of two amyloid and tau radiopharmaceuticals”: it has been added to distinguish the studies concerning the clearance and dosimetric aspects.
A better explanation of the various proteins cited in part 3.6 has been added.
In the introduction of paragraph 3.6, it has been explained how these ligands have been discussed as follows: “As said before, AD has in common with tauopathies the presence of intraneuronal tau inclusions [9]. Therefore, although AD is considered as a secondary tauopathy, it could be interesting to cite other thrilling advances in neuroimaging in cognitive decline from Alzheimer’s disease.”
The English language in the manuscript has been revised and grammar mistakes and typos have been corrected.
Reviewer 2 Report
This is a review article ostensibly about external brain imaging of misfolded tau protein, which is present in a variety of neurodegenerative diseases. The clinical and scientific communities would benefit from a critical review of this important area, However, I have serious concerns about the way this paper is structured, such that much of the important information in the cited studies is not presented well. My concerns are:
1. The English frequently has relatively minor errors, mainly subject-verb disconnection, that is easily corrected by a native English speaker.
2. Much more importantly, the findings of the many studies reviewed are presented in a telegraphic manner, with one immediately following the other, such that the specific findings of each study (while accurately presented) are lost to the reader. Paragraphs or use of bullet points would help. There has to be a better way to present the large volume of data emerging from the authors' review. I suggest some kind of summary Table that addresses the central question (Tau imaging), perhaps presented as animal studies or human studies, and with other imaging modalities noted (e.g., beta amyloid PET, MRI, TPSO receptor, etc).
3. The authors understandably drift into related areas (novel ligands for imaging beta amyloid subspecies, neuroinflammation, etc.) While interesting, these areas are not the primary stated focus of the review, which is imaging misfolded Tau isoforms.
4. The references are at times incorrect. The authors need to review their cited references to be certain that they are correct. As an example, the authors refer to "Meier, et al" (line 262). In fact, the correct reference is Syvanen, et al Neuroimage, March 2017 1; 148:55-63, doi 10/1016. j.neuroimage,2017.01.004. Epub 2017 Jan 6. There are typo errors in their discussion of this study, such that they refer to "16 y.o. mice" (line 268, y.o.=year old) when in fact the mice were 16-17 months old. This error is repeated on line 272.
Overall, this is potentially a valuable contribution. The authors have clearly done a significant amount of discriminating literature review, but now need to modify their presentation to the benefit of the reader.
Please see above. This could be a major contribution to the field but needs attention to minor English errors, major attention to presentation, and checking of references.
Author Response
First of all we would like to thank you for the attention shown and the precious suggestions made, which we have welcomed. We have been glad to put them in place, conscious of the fact that your advice and suggestions will certainly improve the quality and comprehensibility of our work. All recommended corrections have been made and underlined in the manuscript, as detailed below.
- The English language in the manuscript has been revised and grammar mistakes and typos have been corrected.
- The paper has been ameliorated and a table summarizing the considered studies has been added (tab 1). The subparagraph “3.6.1 Clearance and biodistribution studies of two amyloid and tau radiopharmaceuticals” has been added to distinguish the studies concerning the clearance and dosimetric aspects.
- In the introduction of paragraph 3.6 it has been explained how these novel ligands have been discussed as follows: “As said before, AD has in common with tauopathies the presence of intraneuronal tau inclusions [9]. Therefore, although AD is considered as a secondary tauopathy, it could be interesting to cite other thrilling advances in neuroimaging in cognitive decline from Alzheimer’s disease.”
- Ref 51 in line 262 has been changed in ref 52 referred to “Meier SR, Sehlin D, Roshanbin S, Falk VL, Saito T, Saido TC, Neumann U, Rokka J, Eriksson J, Syvänen S. 11C-PiB and 124I-Antibody PET Provide Differing Estimates of Brain Amyloid-β After Therapeutic Intervention. J Nucl Med. 2022 Feb;63(2):302-309. doi: 10.2967/jnumed.121.262083. Epub 2021 Jun 4. PMID: 34088777; PMCID: PMC8805773”.
All the bibliography has been modified from reference 51.
The reference “16 y.o. mice" and “8 yo” have been corrected with “16 month-old” and “8 month-old” respectively.
Round 2
Reviewer 1 Report
One question from the previous comments was not addressed (item #12 below) in this revision and there is more discussion (item #18 below) should be included. And I strongly recommend the authors to go over the manuscript carefully to check the mistakes and get some professional help to polish the writing. Some of the issues are listed below.
1. Capable to add, should be “capable of adding” or “able to add” in line 127
2. In line 128, with the TSPO-specific radiotracer, PET can distinguish the high and mixed affinity binder, but not necessarily to improve the quality of the image. Please rewrite the sentence.
3. Remove “behaviour” in line 140
4. Please rewrite “Studied in another study” in line 158
5. “behavioural variant frontotemporal dementia” can be abbreviated as “bvFTD” in line 160
6. “Colony stimulating factor 1 receptor” can be abbreviated as “CSF1R” in line 195
7. The meaning of the sentence in line 201-203 was not clear
8. 18F-AV1451 should be placed before 11C-PK11195 in line 214 as the previous sentence describing the correlation between tau aggregation (which is AV1451) and microglia activation (which is PK11195)
9. In line 217, BP should be in italic and ND should be in subscript
10. Please replace figure 2 with a higher resolution image, the current one is barely legible
11. “Even if” in line 282 should be “while”?
12. The author did not answer how the comparative study on amyloid tracers can benefit the understanding of Tau pathology in section 3.4, instead a brief comparison among the tracers discussed in this section was added.
13. “are high value” should be “are of high value” or “are highly valuable” in line 324
14. “The results assessed” should be “the study assessed” or “the results indicated” in line 330
15. Line 326-330 should be removed as it is same as what’s in the ref 56, and replaced with “The same group continued to compare the uptake in several regions with more details and found…”
16. What’s the author trying to say with “ulterior” in line 343?
17. “Intracranial” is not exactly interchangeable with “intracerebral” in line 348
18. It would be better to re-categorize section 3.6 as “novel radiotracers to image cognitive decline”, and in this section, the discussed tracers are not the first PET radiotracer for its target, please discuss how is the new tracer comparing to the existing tracers for the same target.
19. Ref 74 did not discuss anything relevant to facilitating the blood-brain barrier penetration, why the author added one sentence about the BBB penetration in the beginning of the paragraph in line 479?
20. “Their” in line 488 refers to what?
One question from the previous comments was not addressed (item #12 below) in this revision and there is more discussion (item #18 below) should be included. And I strongly recommend the authors to go over the manuscript carefully to check the mistakes and get some professional help to polish the writing. Some of the issues are listed below.
1. Capable to add, should be “capable of adding” or “able to add” in line 127
2. In line 128, with the TSPO-specific radiotracer, PET can distinguish the high and mixed affinity binder, but not necessarily to improve the quality of the image. Please rewrite the sentence.
3. Remove “behaviour” in line 140
4. Please rewrite “Studied in another study” in line 158
5. “behavioural variant frontotemporal dementia” can be abbreviated as “bvFTD” in line 160
6. “Colony stimulating factor 1 receptor” can be abbreviated as “CSF1R” in line 195
7. The meaning of the sentence in line 201-203 was not clear
8. 18F-AV1451 should be placed before 11C-PK11195 in line 214 as the previous sentence describing the correlation between tau aggregation (which is AV1451) and microglia activation (which is PK11195)
9. In line 217, BP should be in italic and ND should be in subscript
10. Please replace figure 2 with a higher resolution image, the current one is barely legible
11. “Even if” in line 282 should be “while”?
12. The author did not answer how the comparative study on amyloid tracers can benefit the understanding of Tau pathology in section 3.4, instead a brief comparison among the tracers discussed in this section was added.
13. “are high value” should be “are of high value” or “are highly valuable” in line 324
14. “The results assessed” should be “the study assessed” or “the results indicated” in line 330
15. Line 326-330 should be removed as it is same as what’s in the ref 56, and replaced with “The same group continued to compare the uptake in several regions with more details and found…”
16. What’s the author trying to say with “ulterior” in line 343?
17. “Intracranial” is not exactly interchangeable with “intracerebral” in line 348
18. It would be better to re-categorize section 3.6 as “novel radiotracers to image cognitive decline”, and in this section, the discussed tracers are not the first PET radiotracer for its target, please discuss how is the new tracer comparing to the existing tracers for the same target.
19. Ref 74 did not discuss anything relevant to facilitating the blood-brain barrier penetration, why the author added one sentence about the BBB penetration in the beginning of the paragraph in line 479?
20. “Their” in line 488 refers to what?
Reviewer 2 Report
This is a revision of a paper I previously reviewed. The paper has been significantly improved. Specifically, the authors have added a summary sentence to the end of most paragraphs, and these summary sentences will be very helpful to the readers. Second, the authors have added a summary table (Table 2) that succinctly summarizes all the radioligands used in their article.
The minor English problems appear to have been corrected.
The bibliography (so important for review articles) has been corrected.
Author Response
Let me reiterate my thanks to the revisor for the precious comments: they were indispensable to ameliorate our work and to understand on what it was necessary to focus.